# Sophisticated Fowl: The Complex Behaviour and Cognitive Skills of Chickens and Red Junglefowl

**DOI:** 10.3390/bs8010013

**Published:** 2018-01-17

**Authors:** Laura Garnham, Hanne Løvlie

**Affiliations:** IFM Biology, Linköping University, 581 83 Linköping, Sweden; laura.garnham@liu.se

**Keywords:** animal behaviour, animal cognition, animal welfare, *Gallus gallus*

## Abstract

The world’s most numerous bird, the domestic chicken, and their wild ancestor, the red junglefowl, have long been used as model species for animal behaviour research. Recently, this research has advanced our understanding of the social behaviour, personality, and cognition of fowl, and demonstrated their sophisticated behaviour and cognitive skills. Here, we overview some of this research, starting with describing research investigating the well-developed senses of fowl, before presenting how socially and cognitively complex they can be. The realisation that domestic chickens, our most abundant production animal, are behaviourally and cognitively sophisticated should encourage an increase in general appraise and fascination towards them. In turn, this should inspire increased use of them as both research and hobby animals, as well as improvements in their unfortunately often poor welfare.

## 1. Introduction

The chicken (*Gallus gallus domesticus*) was domesticated over 8000 years ago (possibly as early as 58,000 ± 16,000 years ago [1]). This domestication took place in South-East Asia [2] and had multiple origins [3]. The red junglefowl (*Gallus gallus*) is the main ancestor of today’s chickens [4,5], although there has been introgression from other junglefowl species, such as the grey junglefowl (*Gallus sonneratii*) [5]. Domestic chickens and red junglefowl are still the same species and interbreed when able. This species is historically and contemporarily a consistently important study species for research on animal biology. While red junglefowl are now endangered, the chicken, due to its success as a production animal, is our most abundant bird. Indeed, they are one of our most important domestic animals, with around 50 billion produced yearly in the world (reviewed in [6]). Unfortunately, chickens also face severe welfare issues under production settings, such as feather pecking and cannibalism (discussed in [7]). Some of these problems may arise due to that the wild red junglefowl is selected for a life in the jungle, in small groups of mixed sexes [8]; this is a life very different from that of today’s commercial flocks of 10,000s of indoor-kept industrialised chickens. Improved understanding of the natural needs, capabilities, and behavioural responses of fowl may help us to provide good welfare for them, under production, research, and hobby conditions. This understanding can be improved by studying captive populations of red junglefowl, or domestic fowl living under more natural conditions (Figure 1).

Overall, studies show that the senses and behaviours of chickens are similar to those of red junglefowl, although the frequency of behaviours may vary. In addition, selection during domestication (mainly for growth and fecundity) does not seem to have affected the cognitive abilities of chickens [9]. This means that they have also remained similar to red junglefowl in terms of cognition (i.e., the process by which they perceive, store and act on, environmental stimuli [10]). Due to these similarities between chickens and junglefowl, we here present the results from research on both interchangeably, and often refer to them collectively as fowl. 

Fowl make good subjects for research for similar reasons that they make good production animals; they are social, relatively easy to keep and habituate to human presence. They have long been used as model organisms for vertebrate development (e.g., [11]) and genetics (e.g., [12]). They have also attracted a long history of research investigating their behaviour (e.g., the formation of their social hierarchies e.g., [13,14]) and, increasingly, their sensory and cognitive abilities [7,9] (Figure 2). 

Unfortunately, humans still generally view fowl simply as a food or a commodity, and therefore perceive them as lacking most of the characteristics used to describe other cognitively advanced animals [9]. However, interacting with fowl and consequently realising that they can show boredom, frustration, and happiness, can cause people to develop a more positive opinion of them, within a matter of just hours [15].

In contrast to the general view, there are several recent extensive reviews that cover the sophisticated behaviour (e.g., social and sexual behaviour [6,7]) and well-developed cognition (e.g., [7,9]) of fowl. We here aim to summarise these reviews briefly, and expand on them by including recent findings on animal personality (i.e., consistent between-individual variation in behaviour [16]), affective state (i.e., an individual’s moods and emotions [17]) and cognition. In doing so, we aim to improve the overall view of the fowl, and in turn, inspire not only improvements in their welfare, but also an increased interest in their use both for scientific research and as hobby animals. 

We begin with an overview of the well-developed senses and complex social behaviour of fowl, which may explain some of the sophisticated behaviour and cognitive performances observed. We then discuss the personality, affective state, and cognition of fowl. Where relevant, we briefly discuss the potential implications of findings for their welfare.

## 2. The Sensory Abilities of Fowl

Fowl possess well-developed visual, auditory, olfactory, and tactile senses (also discussed in [7]), as well as potentially magnetoreception [18]. Regarding their visual ability, fowl use each eye for a different purpose; the right eye for smaller details (such as food [19]), and the left eye for novel stimuli, predators [20], and distinguishing familiar from unfamiliar conspecifics [21]. This would presumably be a useful adaptation for fowl as ground dwelling birds, as it enables them to divide their attention between looking out for predators and searching for often hidden food. Fowl possess good short and long-distance vision [22,23], complemented by excellent detection of a broad range of colours, including ultraviolet (UV) [7], which is thought to be used for rapid movement detection [24]. Fowl can detect higher flicker frequencies than humans (up to 100 Hz, depending on light intensity [25]) and may perceive artificial light as flickering if this has lower flicker frequencies than natural light. This, and that artificial light often lacks UV [7], may limit the visual abilities of fowl housed in artificial light. These aspects of artificial light could potentially result in behavioural changes, especially if the light intensity is very low (e.g., around one lux), as some aspects of social behaviour are perturbed at these levels [26].

The overall, traditional, view is that birds have poorly-developed senses of smell (olfaction) and taste (gustation), which they rarely use (however, see e.g., [27]). Contrary to this, these senses are well developed in fowl [7]. Even on the day before hatching, chicks can detect and react to olfactory cues (reviewed by [28]). Fowl use olfaction in numerous behavioural contexts [28], including predator awareness. They appear to act more fearfully towards predator odours [29], and it is suggested that they produce olfactory cues in response to predators that can act as a warning signal to their conspecifics [28]. Understanding the olfactory abilities of fowl could be used to improve their welfare, for example, if fowl are exposed to familiar scents in novel situations this seem to decrease how fearful they are in these situations [30].

Chicks gain the ability to hear around incubation day 12 [31]. They can therefore communicate with each other before hatching and, in doing so, may synchronise their hatching times [32]. Fowl can detect sound frequencies from as low as below 20 Hz [33] to as high as 4000 Hz [34]. This means they can perceive infrasound [33], below the lower limits of human hearing. Lower frequency sounds play an important role in female-chick communication [7] and low-frequency, repetitive sounds have been found to aid the imprinting process [35]. High noise levels (e.g., 80 db) have been found to reduce egg laying and cause behavioural changes [36], which implies that loud sounds could negatively affect welfare. 

Fowl possess a good tactile sense, particularly in their beak, which they use to manipulate items (see references in [7]). Beak trimming, used to avoid feather pecking, is common practice in the poultry industry [37]. However, as the beak contains numerous nerve endings, this is likely both painful and stressful [37]. This has led some countries to implement bans on beak trimming. This is an example of how awareness of the sensory system of chickens is used with the aim to improve their welfare.

Fowl may possess magnetoreception as well. Their beaks contain structures (iron-containing sensory dendrites) similar to those found in homing pigeons [38] (a species known to detect and navigate by magnetic fields [39]). Beak trimming decreased the ability of chicks to find hidden food using a magnetic stimulus [38], indicating that these structures are involved in magnetoreception. Fowl also appear to use a magnetic compass, this may help them to orientate within their home-range, allowing fast and efficient movement between important sites (e.g., foraging and roosting sites [40]). 

These well-developed senses enable fowl to gather detailed information about their environment, which in turn can facilitate their sophisticated behaviour and cognition. To provide good welfare, the environment fowl are kept in should carefully be considered, with the aim to enable them to make full use of, and perform natural behaviours, based on these senses. 

## 3. The Social Life of Fowl

The complex social life of fowl serves as a good example of their behavioural sophistication and is probably intricately linked to their impressive cognitive abilities. Under natural conditions, both domestic fowl and red junglefowl show a range of social structures, but often form groups of 2–15 individuals with a slightly female biased sex ratio (e.g., [6,8,41,42]). These groups live within a territory defended by the dominant male, however, interactions between groups can occur and individuals can move between groups [8,42].

Starting from around five to six weeks of ages (around the age chicks naturally become independent from their mother [42]), chicks start to form social hierarchies [43]. These hierarchies are relatively stable and sex-specific, can be linear or more complex [44], with sexually-mature males ranking higher than females overall [43]. Several factors are involved in whether an individual will become dominant, such as morphology, where male comb size is positively correlated to dominance [45]. Behavioural factors are also involved, with males that explore faster, are more aggressive, or more vigilant after a startle being more likely to become dominant [46]. Social status is important, as it influences an individual’s access to resources, which typically positively correlates with reproductive success (males, reviewed in e.g., [6,47]; females [48]). While more dominant males have more reproductive encounters than subordinate males, dominant females can be more reluctant to mate than subordinate females [43]. Social status also, particularly in males, affects frequency of behaviour; dominant males are in general more active, explorative, vigilant, and crow more than subdominant males (e.g., [49,50]). The stability of the hierarchy itself can have implications for the individuals in the hierarchy, regardless of their social status. This is because individuals living in a stable hierarchy are less aggressive toward each other, eat more food, and lay more eggs than individuals exposed to social flux [51]. Keepers of fowl should consider these findings and the negative impact on welfare that reorganisation of their flock, or the introduction of new individuals, could have. 

An important aspect of the behaviour of fowl is that both sexes are promiscuous [6,41,47]. Males aim to both attract females and repel other males, while females aim to mate solely with the males they prefer, and avoid other males’ mating attempts (reviewed in [6,41,47,52]). In red junglefowl, younger and more aggressive males tend to mate more often and with the highest numbers of females [53]. Females also prefer to mate with males that are not related to other males in the group, which in turn could increase genetic diversity and avoid inbreeding [54]. Females prefer to mate with males that provide the most food through courtship feeding, regardless of the social rank of the male [55]. Males, therefore, perform such food displays to attract females who, in turn, eavesdrop on these displays [56]. As the vocal signals of fowl are individually distinctive, they can be used for individual recognition [57]. Females can thus discriminate males by their food calls and males may gain a ‘reputation’ in terms of their food provisioning [58], and females may avoid males that tend to food display without food [56]. 

Dominant males often enjoy higher paternity than subdominant males (reviewed by [6,47]), most likely because they, to some extent successfully, aim to monopolise access to females, re-mate more often with females, and are preferred by females. This female preference for dominant males is shown both directly (reviewed by [6,47]), and indirectly [47,59,60]. An example of the latter is that females distress call during copulation by low-ranking (i.e., non-preferred males) [60]. In doing so, they attract higher ranking (preferred) males, which can disrupt the copulation and then mate with the female themselves [59,60]. However, due to post-copulatory sexual selection, high mating success may not translate into paternity (reviewed by [6,47,61]. Female fowl can store sperm for up to two weeks (reviewed in [41,62]) and have multiple sexual partners [47,63], resulting in intense sperm competition [6,47]. Both males and females have developed sophisticated responses to contend with this strong selection pressure. Males allocate sperm differently between females depending on their previous mating history with these, their fecundity [61], and genetic relatedness to the male [64,65]. Females exert cryptic choice and bias the outcome of mating against subdominant males by ejecting their ejaculates [66] and bias sperm use against genetically similar males after insemination [65,67]. 

The sensory abilities of fowl play an important role in their social (both sexual and non-sexual) lives. Their well-developed visual and auditory senses make it possible for them to have a complex communication system (involving a repertoire of around 24 vocalisations [68]), as well as different visual displays, such as those used during the establishment of social status [69]. Further, in some contexts (e.g., foraging) individuals prefer to associate with others that they are familiar with [22]. This is facilitated by their abilities to discriminate between conspecifics, not only vocally [56], but also visually [70], and potentially olfactorily [30,71]. Visual individual recognition is thought to be by facial recognition [72,73], used by both chicks and adults (e.g., to guide social interactions [74] and mate choice [61,75]). Fowl may also be able to recognise individuals by olfaction. Direct support of this is lacking, but they possess individual body odours [71], and respond to familiar odours [30], which suggests that individual recognition could be the case. Fowl may be able to determine genetic relatedness by olfaction, thus enabling kin recognition (which does not appear to be based on social familiarity, per se [67]). In support of this, genetic relatedness among individuals has several effects on social behaviour. This includes interactions among males and females in a sexual context [53,54,65,67], and related males being less aggressive towards each other, compared to unrelated males, in pre-copulatory competitions over copulations [54,76]. However, sperm competition between related males was more intense than that between unrelated males [54]. This may be because post-copulatory competition has less risk of injury than pre-copulatory competition, reducing the cost of competition among relatives [54]. 

The complex social and sexual lives of fowl can have important implications for their welfare when they are kept under non-natural conditions. An understanding of fowl social behaviour could be used to predict where conflicts among individuals could arise and take appropriate action to prevent these from happening. One such conflict that could occur is sexual conflict between males and females over mating opportunities. This conflict can arise because males sexually harass females in a commercial setting [6,41,53]. This has been shown to cause suboptimal feeding and space use by females [77]. However, female welfare can be improved by providing wood panels so that females can be out of sight of males [77,78]. The presence of males can positively improve female survival rates [79]. Thus, finding a way to successfully house hens with roosters could have a positive impact on female welfare.

## 4. The Personality of Fowl

Animal personality has been described in fowl, such as exploration, activity, aggression, neophobia, and fearfulness [46,50,80,81,82]. This demonstrates that fowl can be individually unique in their behaviour. Personality in fowl shows some degree of heritability, the degree of which depends on the trait being tested [83], suggesting some scope for selection. Personality is, nevertheless, affected by ontogeny and individuals appear more consistent in their responses to personality assays if assayed before eight weeks of age, and after sexual maturity (older than six months of age [84]), than between these time periods [80]. There also appears to be at least a short-term interplay between social status and personality [50,85]. A number of personality traits, including aggression, can influence the establishment of social status among same sex individuals [46,86] and the behavioural response of male fowl in personality assays can be influenced by their social status [50]. Recent experience of victory in a social competition can also make males more aggressive in future intra-sexual interactions [86]. Individual variation in personality and behaviour can, thus, clearly influence how fowl interact with their world, which can have consequences for their social life.

An individual’s personality may have important implications for welfare. Hens with a more reactive personality appear to find acute stressors more stressful than those with a more proactive personality type [87]. This indicates that certain personality types may be better able to cope with stressors than others. In addition, certain personality types may be more prone to display unwanted behaviour. For instance, hens with a more proactive personality were more likely to feather peck than those with a more reactive personality (reviewed in [88]). Knowing which personality types are associated with potential welfare issues can enable focused attention on these individuals and faster noticing problems if they occur. Individuals may also differ individually in what they require to experience good welfare. Therefore, designing their set up so that fowl can make choices about what they experience, and so have increased control over their situation, may help reduce stress and provide better welfare [89].

Further, the interaction between personality and social behaviour could have welfare implications, particularly in situations in which fowl are housed in large groups within limited space. Such housing increases the potential for aggressive interactions, both to occur and to be repeated (as fowl may have difficulty visually discriminating individuals in large groups [90]). This situation may be made worse by domestication, as, opposite to the pattern usually observed in domestication; certain breeds of domestic fowl seem to be more aggressive than wild-type fowl or red junglefowl [91]. Aggression may be reduced by avoiding frequent regrouping as aggression is lower when a stable hierarchy is formed [51], by adding visual barriers [77,78], or by housing related individuals together [76]. Finally, it is important to be aware of birds acting abnormally, as this has been found to attract aggression [7].

## 5. Affective State in Fowl

Both the social behaviour and cognition of fowl can be affected by their affective state. Regarding social behaviour, affective state appears to play a role in the establishment of social hierarchies [14]. Hens that had recently experienced victory, and were, therefore, presumably in a positive affective state, were more likely to become dominant in a new group [14]. Regarding cognition, increased fear correlates with poorer performance in cognitive tests in fowl [92]. Good welfare practices should aim to increase positive affective states and reduce negative affective states. Yet, welfare often tends to focus on reducing negative affective states (reviewed in [7]). Doing so is important as negative affective states can have adverse effects on health, social interaction, and the ability to cope with change [93]. Examples of how welfare can be improved to reduce negative affective state include implementing machine rather than manual catching, which can reduce fear and stress [94]. Further, providing environmental enrichment can reduce fearfulness [95] and stress-induced negative judgement bias [96]. 

More recently, there has been interest in increasing positive affective state, thought to be just as important for welfare as reducing negative affective state [97]. The affective state, both positive and negative, of individuals can be assessed by observing their response to ambiguous cues intermediate between cues with known positive and negative values (i.e., cognitive judgement bias test [98]). Individuals considered to be in a negative affective state respond to ambiguous cues more pessimistically (e.g., [99]) and those considered to be in a positive affective state respond to ambiguous cues more optimistically (e.g., [96,100]). This approach has shown that providing environmental enrichment can keep individuals in a positive affective state even when exposed to additional stress [96]. Additionally, if an object chicks have imprinted on is included in their housing, they can show a more positive affective state (i.e., they do less distress calling [101] and respond less aversely to stressful experiences [102]). This indicates that imprinting could be used to improve positive welfare.

Furthermore, fowl not only possess their own affective states, but also appear to be aware of the affective states of their companions. They show a greater stress response to their companions being handled roughly than to them being handled gently [103]. This suggests that fowl can experience emotional contagion, where the emotions of one individual can trigger similar emotions in observers (reviewed by [9]), which indicates that empathy could be possible in this species [7,9]. Therefore, fowl do not have to directly experience a stressful situation to become stressed, but instead could become stressed as a result of the individuals they are housed with becoming stressed.

## 6. Cognitive Abilities in Fowl

There is increasing interest in the cognitive abilities of fowl (e.g., [7,9]). Chicks can be imprinted on objects other than their mother (imprinting is the phenomena in which chicks memorise the properties of the first moving object they see and, afterwards, show a preference for this object, reviewed by [103]). Some cognitive tests make use of this, as chicks have a desire to reunite with imprinted objects when separated from them (e.g., [104,105]). Imprinting differs from general learning and memory in that is has higher learning efficiency and more robust memory retention [103]. Imprinting has been used to investigate other cognitive processes, such as perceptual learning and generalisation. Whether a chick imprints on a stimulus depends on whether they categorise that stimulus as social or non-social and chicks use perceptual learning to perform this categorisation and imprint selectively on social stimuli [35].

Social learning plays an important role in the lives of fowl [7,9]. This includes passive avoidance learning, in which fowl learn what to avoid by observing the experiences of others [106]. Interestingly, fowl appear to learn better from trained demonstrators [107], and dominant individuals (perhaps because they naturally pay more attention to these [108,109]). Social learning could explain why fowl sometimes synchronise behaviours [110], and, to enable good welfare, there should be sufficient resources to enable this. Undesirable behaviours, such as cannibalism, can also spread by social learning [111], however, this could be prevented by increasing space and providing visual barriers.

Transitive inference entails inferring relationships between items that have not been directly compared before (reviewed in [112]). Fowl use this to navigate both social hierarchies [113] and abstract comparisons [114]. This can enable them to avoid competitions that they are likely to lose. For example, when hens observed the outcome of a duel between a novel hen and a familiar, dominant hen, they only entered an aggressive encounter with the novel hen when the dominant hen won [113].

The complex communication system of fowl demonstrates a range of impressive cognitive abilities (reviewed in [7,9]). To begin with, fowl adjust their calls based on their audience: males are more likely to alarm call in the presence of familiar conspecifics [115], and hens only do so for small hawks when their chicks themselves are small [116]. Females give distress calls that attract males who interrupt copulation only when a dominant male is present to do so [59]. Subordinate males produce quieter food displays when dominant males are present [117]. Thus, fowl communication appears to be volitional, utilise social awareness, and involve perspective-taking (reviewed in [9]). This is an ability associated with some of the most advanced forms of cognition, including theory of mind [118]. Hens also appear to take the perspective of their chicks, giving stronger maternal displays, and becoming more stressed [119,120], when their chicks are in situations they themselves previously experienced as unpleasant [121]. Fowl also adjust their calls depending on the subject matter. They (usually dominant males) produce different alarm calls for terrestrial than for aerial predators (reviewed in [7]). Males produce more vigorous food calls for better quality food [122]. Receivers can respond appropriately to these calls, presumably by creating mental representations of the subject without needing direct experience of this (reviewed in [9]). Mental representations enable advanced cognitive abilities, such as understanding object permanence, and perceptually filling in hidden sections of objects [104]. As the same call is consistently used for the same subject and invokes the same response, this implies that fowl use referential communication. This requires advanced cognitive abilities, previously attributed only to certain primates (reviewed in [9]). Fowl communication also shows evidence of risk assessment and deception, both thought to be cognitively advanced behaviour [123]. In terms of risk-taking, males produce more alarm calls when closer to cover [124], and/or in the presence of a female [125]. In the latter case, he may offset the risk to his own survival with the potential inclusive fitness gains by offspring survival [125]. In terms of deception, males use food calls to attract females when there is no food available [126]. 

Sophisticated cognitive abilities are also used by fowl in non-social contexts. Fowl show the self-control needed to wait for larger delayed reward instead of going for the immediate gratification of an instant small reward [127]. In humans, self-control positively correlates with cognitive ability [128]. Being able to show self-control suggests the possession of episodic memory which indicates that fowl experience life autobiographically and can mentally place themselves in the past or future, or alternate situations [9]. This, in turn, suggests that they possess self-awareness (reviewed in [9]). Fowl also appear to use sophisticated trade-off calculations between time and reward size, which humans can only manage past four years of age [129], to decide whether it is worthwhile to wait [127]. When the reward size was increased from close in size to the instant reward to a much larger ‘jackpot’, the number of tested hens that waited for the larger reward increased from 22% to 93% [127]. 

The memory of fowl is as good as most primates (reviewed in [9]), and in some cases better (e.g., when remembering the location of certain biologically-attractive objects [130]). Young chicks can remember a detour taken 24 h earlier [131], and remember the location of stationary, and predict the location of moving, hidden objects after a delay of 180s [105]. As adults, memories can last over years [personal observation], this could mean that fowl may remember stressful experiences long after the experience has ended and could potentially suffer negative welfare because of this. Fowl appear to store information in the form of declarative representations [132]. These contain general information about the relationship between events that can be applied to other situations [132]. Hens, for instance, appear to create mental representations of what is edible, which they can use to correct their chicks’ foraging choices [119].

Regarding mathematical abilities, newly-hatched chicks can distinguish between, at least, small quantities and sequence numbers [133,134,135]. Interestingly, fowl seem to map numbers on to geometrical space using a mental number line, a method similar to that often used by humans [134]. They also understand some concepts of physics, including structural engineering and time, to some degree. In terms of the former, chicks can distinguish between possible and impossible three-dimensional structures [136], and encode information about relative length and angles [137]. In terms of the latter, hens can predict six-minute intervals [138].

As can be seen, fowl show a range of sophisticated cognitive abilities in both social and non-social contexts. One of the reasons fowl do well at certain cognitive tasks might be that they possess a lateralised brain, that is, the left and right hemispheres of their brain are specialised for different purposes [139]. This is thought to enhance brain efficiency in cognitive tasks that require the use of both hemispheres at the same time for different purposes [140]. Chicks with lateralised brains can simultaneously attend to two different visual inputs (e.g., discriminate grain from pebbles, and detect a predator moving overhead at the same time), whereas chicks with reduced brain lateralisation cannot [140]. The less lateralised chicks took much longer to detect a predator model and once they had done so, were so distracted they could no longer successfully distinguish food from pebbles [140]. Lateralisation can also improve performance in more complex cognitive processes, such as representation learning and transitive inference [114]. 

Importantly, fowl show individual differences in terms of cognitive performance with a large variation in learning speed being observed for both chick and adult red junglefowl, in a range of learning tasks [141]. Interestingly, individuals differ in terms of how fast they learn different tasks [141] suggesting that there is no overall “smarter” fowl. Several factors could be responsible for these individual differences in learning speed. Performance in cognitive tests can vary depending on the age and sex of individuals; female and male fowl both show improvements in maze navigation with age, and males perform better at this than females overall [142]. Personality can also affect performance on cognitive tests, for example, domestic hens with a more reactive coping style performed better on associative learning tests compared to more proactive hens [143] and exploration explained variation in learning speed in both chicks and adult red junglefowl [141]. In addition, cognitive stimulation seems to affect personality later in life [81]. Individuals that took part in a variety of cognitive tests as chicks were found to become more vigilant and less likely to attempt to escape in later personality assays, compared to chicks that did not experience this cognitive stimulation [81]. Learning speed can also depend on the properties of the cues used, with novelty, salience, and the scent of pyrazine all being properties that make avoidance learning easier for fowl (reviewed in [7]). Additionally, fowl seem to do well on tasks that tap into their social propensities (reviewed in [9]), and certain experiences as chicks, for instance environmental stimulation, can improve learning later in life [144].

## 7. Conclusions

We have here reviewed the sensory abilities, social and sexual behaviour, personality, affective state, and cognition of fowl, including, where relevant, some of the implications of these topics for their welfare. Knowledge of the impressive sensory abilities of fowl can help design housing that enables fowl to use the full range of these abilities. As these abilities can surpass those of humans, it is important to bear in mind that conditions that are acceptable to humans may not be so for fowl. Fowl show sophisticated social behaviours, in both sexual and non-sexual contexts. Understanding these behaviours can help provide fowl with good welfare, for example, by ensuring that fowl have the resources they require to perform these behaviours, or by predicting and reducing welfare issues that could arise because of them. That fowl display personality shows they can also be individually unique in their behaviour. Thus, certain individuals may be more prone to welfare problems, or have different requirements to experience good welfare. Due to this, welfare should be focused at the level of the individual. Measuring the affective state of fowl could be used to assess their welfare and determine actions that could improve this by increasing positive affective state and decreasing negative affective state. That fowl experience emotional contagion suggests that that their affective state may not only be affected by their own experiences, but also the experiences of those around them. Fowl display a range of impressive cognitive abilities, disproving the notion that they are cognitively inferior. Thus, welfare practices once thought acceptable may need to be reconsidered. This is because fowl may perceive their situation differently than previously thought, and current practices may deprive fowl the cognitive stimulation they require. That fowl already show evidence of multiple complex behaviours and cognition should inspire further research into their cognition to determine what else they are capable of. Overall, an understanding of the behavioural and cognitive sophistication of fowl should encourage a more positive and respectful opinion of them. This should drive improvements in their welfare, regardless of the purpose for which they are kept. Further, this should inspire people to experience and learn more about this fascinating species, both through research and by keeping them for recreation.

## Figures and Tables

**Figure 1 behavsci-08-00013-f001:**
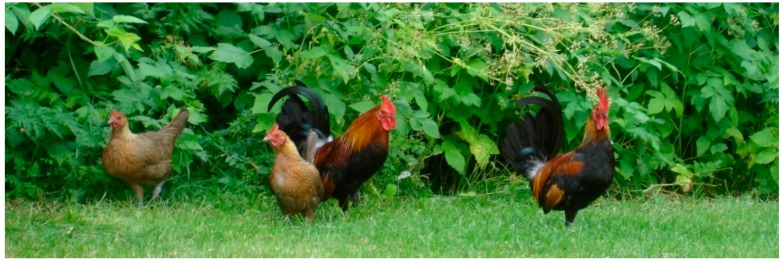
Free-ranging group of a Swedish game breed of domestic fowl.

**Figure 2 behavsci-08-00013-f002:**
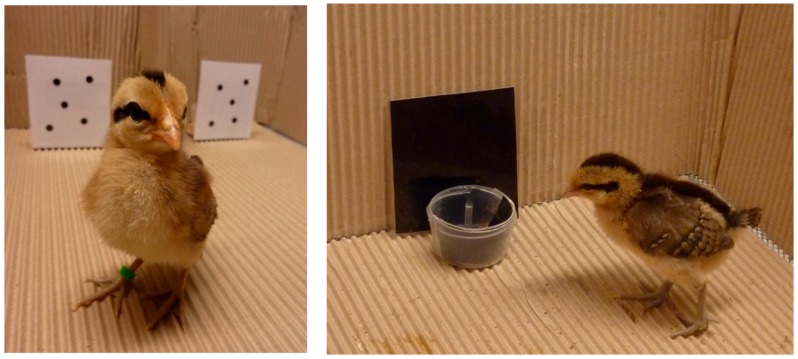
Red junglefowl chicks in behavioural assays measuring cognitive performance.

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
