# Peer review of "Sophisticated Fowl: The Complex Behaviour and Cognitive Skills of Chickens and Red Junglefowl"

_behavsci, 2018, doi:10.3390/bs8010013_

Round 1

Reviewer 1 Report

This a broad reveiw which will interest those new to the field. As far as I can tell all the information is accurate and uptodate and not controversial.  The prose is clear and with very minor excpetions the English is good. 

Author Response

Thank you for your review of our work, please see attached our reply.

Reviewer 2 Report

This paper represents a nice idea, since the issue of the social regulation of domesticated Jungle fowls is highly captivating. More importantly, the resulting reflections in terms of minimising psychophysical suffering under cage constraints is definitely an important one.

However the paper has a minimal bibliography and most of the concepts are only touched upon. Overall, the paper should enlarge the text of 25-30 % of its present length.

Specific points:

Lines 23-24 even earlier: such a genetic  ancestry, yet putative, deserves an entire paragraph.

46-52: the massive literature on fowls social hierarchies, pecking order etc deserves at least a couple of paragraphs.

63-66 :sexual behaviour and especially cognition need further elucidation ( about 3 para)

91-96: bird olfaction, Floriano Papi’s pioneering studies on homing pigeon navigation neeed to be cited, with other authors, F. Bonadonna, seabirds, Annina Gagliardo, etc.

110-120: the impact on welfare is only mentioned; indeed, it represents a core issue of the present paper.

119-120: in particular, male intraspecific aggressions are of paramount importance in terms of managing social settings. An entire para should be devoted to them and how to cope with escalation leading to injuries.

146-156: how selection for domestication more precisely molded sensory “gates” in this strains? Clarify.

169-175: affective state. This is an original perspective, in need of further details and operative refletcions for managing.

184- : fowl personality. Operational indications for keepers should be made by adding more space to them; as in the case of suine subjects (see papers by Francoise Wemensfelder, Edinburgh) animal personalities may in fact conflict with human personalities.

378-389: conlusions. They are definitely very, very poor statements.

Author Response

(The authors gave the same response as above.)

Reviewer 3 Report

This manuscript provides an overview on complex behaviour and cognition in the fowl. The aim of this review is an important one, i.e. to increase the readers' awareness and consideration of the fowl by bringing to attention the huge set of experimental results obtained about this species in the last 30-40 years. This is particularly relevant when considering that the fowl is the most abundant production animal, a species subject to intensive farming and - too often - to poor welfare conditions.

The manuscript is well written (I detected only some minor text editing issues, e.g. line 161 "se" instead of "see"; line 337 understanding "of" the sensory....; line 369 "not depend not") and the evidence reported is correctly and clearly described, although often quite superficially, especially for what concerns the section on sensory ability and on cognition.

My main concern regards the fact that this review in its current state is too similar in structure and content to the recent review published on the same topic by Lori Marino (Animal Cognition, 2017). If a reader has already had the chance to read the Marino's article, he/she would find this reading pretty much of a duplicate. Moreover some of the issues - as mentioned above - are tackled at a more superficial-generic level. The part on sensory abilities is too similar to Marino's paper and it is disappointingly succinte, same holds true for the section on complex cognition. There are notable exceptions though: the parts dealing with sexual selection, personality, and welfare implications.

I therefore reccommend the authors to revise this article and extending those parts by providing more detailed information, and in general to allocate much more attention to the most recent literature, which is not present in Marino's paper.

The domestic chicken has been extensively studied also with regard to topics which are not at all considered in the present paper (nor are they mentioned in Marino's review) such as brain lateralisation, face perception (starting from Morton and Johnson's seminal paper) and the neurobiology of learning and memory (investigating mechanisms of learning such as filial imprinting or passive avoidance). These are just some suggestions for the authors in order to enrich and individuate their paper by broadening sections which do not overlap with the existing general review literature.

Other issues:

Much of the discussion on cognitive abilities (pages 6 and 7) is actually dealing with affective and emotional behaviour rather than proper cognition. I wonder if it would be worth to have a separate section dealing with those issues.

Also, please check the referencing throughout as it seems that there are some mis-quotations (e.g. line 311 ref. 94, and possibly others). References number 73 and 104 are incomplete.

When discussing the implications for welfare, please provide evidence-based advise, or, if claims are not evidence-based this should be clearly stated (e.g. lines 341-42 about artificial light; or lines 351-353 on social learning of feather pecking).

Concerning welfare implications, maybe some information on the actual legislation and on its historical evolution to accomodate scientific evidence on chickens' behaviour would be of some interest to the readers.

line 305: I believe "perceptually filling in" is more correct than the current expression "mentally filling in".

Author Response

(The authors gave the same response as above.)

Round 2

Reviewer 2 Report

THIS REVISED VERSION IS OK NOW.

Reviewer 3 Report

I appreciated the changes made by the Authors in their revised version. The new version is in my opinion much improved and suitable for publication in its present form.